# The Psychological Impact of the COVID-19 Pandemic on Postsecondary Students: An Analysis of Self-Determination

**DOI:** 10.3390/ijerph19148545

**Published:** 2022-07-13

**Authors:** Paige S. Randall, Paula D. Koppel, Sharron L. Docherty, Jennie C. De Gagne

**Affiliations:** School of Nursing, Duke University, 307 Trent Drive, DUMC 3322, Durham, NC 27710, USA; paula.koppel@duke.edu (P.D.K.); sharron.docherty@duke.edu (S.L.D.); jennie.degagne@duke.edu (J.C.D.G.)

**Keywords:** concept analysis, coronavirus pandemic, global health, holistic approach, postsecondary students, psychological distress, self-determination

## Abstract

The COVID-19 pandemic has put postsecondary students across the world at risk of psychological distress, negatively impacting their basic psychological well-being, including self-determination. Although the concept of self-determination has been widely discussed in literature, it is poorly understood within the context of postsecondary students during the COVID-19 pandemic. This study aimed to examine the concept of self-determination (SD) as it relates to postsecondary students amid the COVID-19 pandemic. The Rodgers’ evolutionary method of concept analysis was used. PubMed, CINAHL, PsycINFO, and ERIC were electronically searched using the keywords “postsecondary students” “coronavirus pandemic” and “self-determination.” The historical, legal, educational, and health science literature were investigated to generate a holistic definition of SD in the past. This analysis has identified the antecedents, attributes, and consequences of self-determination in postsecondary students during this global health crisis. This analysis adds to the knowledge base regarding the evolution, significance, and application of the concept of SD in the context of postsecondary students amidst the COVID-19 pandemic. Implications for future research were also explored, such as using strategies to promote SD in postsecondary students to develop resilience during the pandemic.

## 1. Introduction

In March 2020, the World Health Organization (WHO) categorized COVID-19 as a global pandemic due to its effect on individuals all over the world [1]. Since January 2020, there have been over 6,200,000 reported deaths globally [2]. Although this pandemic has had catastrophic effects on the physical health of millions of individuals, the psychological impact of the virus has also proven detrimental [3,4,5,6]. The full psychological impact of the COVID-19 pandemic continues to be investigated; however, research has shown that postsecondary students are a population that is highly susceptible to psychological distress [3,7,8,9]. Young adults between 18 and 24 years of age, the most common age group for postsecondary students, reported the highest levels of pandemic-related psychological distress when compared to other age groups, with a quarter of them reporting suicidal thoughts [10,11].

The COVID-19 outbreak emerged during the 2019–2020 academic year, bringing rapid policy changes for postsecondary institutions across the globe [12]. To slow transmission of the virus, institutions mandated social distancing, virtual classes, and/or quarantine for students who tested positive for COVID-19 or had close contact with someone diagnosed with the virus. Intensified by fear of the virus, such isolation can lead to psychological distress due to chronic loneliness and boredom [13]. Reported symptoms of COVID-19 related to psychological distress in postsecondary students include anxiety, stress, depression, sleep disturbance, uncertainty, and suicidal ideation [7,8,14,15]. When an individual experiences these conditions, it is difficult to satisfy basic psychological needs. One theory highlighting the importance of such needs is the self-determination theory (SDT). Schwinger et al. and Martinek et al. used SDT as a lens to analyze how German adults and postsecondary students, respectively, experienced psychological distress and basic need frustration due to limited self-determination (SD) as a result of pandemic restrictions [5,9]. Therefore, SD can be affected by COVID-19-related academic and social restrictions, putting postsecondary students at risk of psychological distress [5,9].

SD is generally described as the process by which a person controls their own life [16] while the SDT is used to explain how human motivation relates to innate psychological needs. Coined by Ryan and Deci, SDT suggests that personal growth is motivated by one’s universal need for connection or relatedness, competence, and autonomy [17]. When these needs are fulfilled, an individual is described as self-determined. More recently, SDT has been applied to the COVID-19 pandemic with an emphasis on the components of relatedness and autonomy. For example, Schwinger and colleagues used SDT to examine the psychological impact of the COVID-19 lockdown on adults in Germany, finding that adults in lockdown experienced decreased psychological well-being due to COVID-19 restrictions [5]. These government restrictions affected the ability of participants to relate to and socialize with others in addition to limiting their perceived control over their own lives and decisions [5]. SDT is an appropriate lens through which to explore the psychological needs of postsecondary students since environments that do not support SD can foster academic stress and decreased mental functioning [18,19,20], while environments that support SD encourage psychological well-being and resilience [18,21,22,23].

In addition, generational factors of postsecondary students can increase their susceptibility to psychological distress. Many current postsecondary students are members of Generation Z (or Gen Z for short), or those born between 1997 and 2012. When compared with other generations, Gen Z reported more mental health concerns, even prior to the COVID-19 pandemic [24,25]. These concerns may be correlated to Gen Z being “bubble-wrapped” due to helicopter and snowplow parenting [26,27]. These overly controlling parenting styles puts postsecondary students at risk of decreased autonomy, and experience with decision making, thereby limiting their SD and development of coping skills needed to handle stressful situations [22,26].

It is important to consider the evolving nature of SD and apply it to current issues in healthcare, education, and sociology. COVID-19-related psychological distress is a concern for frontline nurses caring for students in behavioral health, school, primary care, and emergency environments. An essential component of the nurse–patient relationship is to assist patients in gaining independence [28]. Nurses can use SD to guide patients’ progress and recovery by recognizing the role patients play in their own healthcare decisions [28]. Although SD has emerged as a central concept in other disciplines, it has yet to be fully analyzed in postsecondary students during the COVID-19 pandemic. For years, SDT has served as a well-accepted framework through which to understand how human beings satisfy basic psychological needs; therefore, the concept of SD may be useful in understand why postsecondary students are facing record high mental health concerns during the pandemic. Hence, the aim of this concept analysis was to explore the concept of SD to achieve a better understanding of the psychological impact of the COVID-19 pandemic on postsecondary students.

## 2. Materials and Methods

A literature search was completed in alignment with Rodgers’ [29,30] evolutionary method of concept analysis. The purpose of the search was to explore how the concept of SD has evolved and to clarify its application to postsecondary students during the COVID-19 pandemic. Rodgers’ [29,30] evolutionary method of concept analysis was chosen for this population since SD has been applied to student populations for decades [31,32], making its application to the COVID-19 pandemic potential useful for understanding the pervasive nature of psychological distress in the postsecondary population. According to Rodgers [29,30], there are three essential aspects of analyzing a concept: (i) significance, (ii) use, and (iii) application. Evaluation of these aspects clarified the meaning of the concept, allowing for further conceptual development within the target population. Although the steps of Rodgers’ [29,30] approach are not always linear, the phases are as follows:(1)identify and name the concept of interest;(2)identify the surrogate terms and relevant uses of the concept;(3)select an appropriate realm for data collection;(4)recognize attributes of the concept;(5)identify the references, antecedents, and consequences of the concept, if possible;(6)identify concepts related to the concept of interest;(7)generate a model case of the concept, if applicable [29] (p. 333).

This Rogerian evolutionary approach to concept analysis is context dependent, which allows for evolution of the concept through various applications. Rodgers states that analysis of a concept “through attention to common use” allows for clarification as the researcher approaches the concept through a new context [29] (p. 333). Therefore, the historical and classic evidence of SD may guide its application to postsecondary students during the COVID-19 pandemic and illuminate the possible etiology and potential interventions to mitigate pandemic-related psychological distress.

The COVID-19 pandemic is evolving every day with emerging implications for many different populations globally. Given the contextual significance of decreasing opportunities for SD in postsecondary students before and during the pandemic, a deeper understanding of the meaning and application of the term SD, through the process of concept analysis, is indicated. Concept analysis allows researchers to continue to develop an idea or issue outside the traditional context of present theory [29,30]. One can successfully examine SD within an evolutionary framework, but it is crucial to select a realm that includes various disciplines and domains to encompass the historical and continued application of this concept [29,30,33].

### Realm (Sample)

Investigation of the meaning of SD within the literature included searching several databases using pertinent keywords. Medline (via PubMed), CINAHL (via EBSCO), PsycINFO (via EBSCO), and Educational Resource Information Center (ERIC) (via EBSCO) were explored. Google Scholar was also used to search for grey literature, including dissertations via ProQuest, government documents, and websites such as the Center for Self-determination Theory [31] and Zarrow Institute on Transition and Self-Determination [34]. The keywords used were “postsecondary students” “coronavirus pandemic” and “self-determination.” The database searches were not limited by time, but due to the recent nature of the COVID-19 pandemic, most literature was published post-March 2020. Several historical government documents were used to explore the evolution of SD, as well as past concept analyses that served as the foundation for SD.

## 3. Results and Analysis

In accordance with the Rogerian method of concept analysis, each source was read to clarify how the concept of SD was defined and used [29,33]. The associated antecedents, attributes, related terms, and consequences were identified as well. The findings from each source were compared to highlight the similarities, which uncovered the changes to the concept over time. Next, a visual map was created to present the linkages between the components of the concept. Finally, the information regarding SD was synthesized to elucidate its application to postsecondary students during the COVID-19 pandemic.

### 3.1. Historical Background

Rodgers’ evolutionary method of concept analysis employs an inductive approach to analysis of a concept within various disciplines over time noted as historical evidence [29,33] Elucidating the historical evidence of a concept lends to its dynamic development and recommendations for future analysis [29,33]. SD is a term that appears within several contexts throughout history. Although healthcare is a more recent vehicle for the application of this concept, it can be viewed through an ethical, legal, political, or societal lens. An early example of political SD is the voyage of the Puritans on the Mayflower in 1620 [35,36]. The Puritans sought freedom and ventured to the United States to find solace from religious oppression, finally settling in a colony at Plymouth [35,36]. The exact term SD was first used in a speech by Woodrow Wilson in 1918 when speaking about territorial conquests during World War I [37].

The notion of SD within the healthcare environment originated in response to the need for patient rights [38]. As medical technology advanced and methods to prolong life became readily available, patients were no longer passive recipients of medical care; instead, they became active players in their own treatment regimen with the passage of the PSDA [35,38]. Within the provisions of the PSDA, patients have the right to make decisions concerning their health, specifically to accept or refuse care [38]. Within several years of the passage of the PSDA, the American Nurses Association (ANA) released an updated edition of The Code of Ethics for Nurses, where the concept of SD heavily influenced nurses in supporting patient decision making in end-of-life care [35,39].

In 1985, Deci and Ryan published a social theory of intrinsic motivation and SD and applied it to social behavior [17]. This theory eventually developed into the self-determination theory (SDT), which includes three key concepts: (i) autonomy (i.e., the perception of control over one’s actions); (ii) competence (i.e., the feeling of mastery when performing a certain skill); and (iii) relatedness (i.e., the sense of social connection with others) [23]. Incorporating these three key concepts is theorized to lead to mental well-being and self-motivation, whereas the lack thereof leads to the opposite effect [17,22,23].

#### Surrogate Terms and Relevant Uses

As defined by the American Psychological Association (APA) dictionary of psychology, self-determination is “the process or result of engaging in behaviors without interference or undue influence from other people or external demands” [40] (para. 1). The Oxford Learner’s Dictionary defines SD as “the right of a country or a region and its people to be independent and to choose their own government and political system” [16] (para. 1). Oxford’s definition can be correlated with the Puritans’ journey for religious freedom aboard the Mayflower; however, the APA’s definition has become more widely used in recent years.

SD has several related terms, including independence, control, and autonomy, which is the most commonly used. The terms autonomy and SD are used interchangeably in the literature [20,35,41]. Although autonomy is a part of SDT [17], it is less a surrogate than a related concept and will be addressed later. SD can be used to examine internal motivation in individuals throughout many different disciplines, cultures, and conditions [31]. For example, SD has been researched in relation to addiction treatment, weight loss, diabetic management, education, occupational therapy, business, athletics, religion, and psychology [22,31,42]. Despite the different applications of SD, what remains constant is the idea that meaningful life choices occur because of internal motivation, rather than external motivation [22], which places the person at the center of their health.

In addition to being used as theoretical framework for many studies, several empirical tools have stemmed from SD [43,44,45]. SD assessment tools have also been used in several studies to measure the ways in which participants act in a self-determined manner [34]. These tools have been used primarily in educational and psychological settings [34,46].

For nurses practicing holistic or integrative nursing, SD and its role in health and healing align with their belief in the interconnected nature of the mind–body–spirit as well as their salutogenic framework of health promotion to support human flourishing and wellbeing [47,48,49]. Person-centered care is an example of a holistic nursing practice model focused on patient preferences, with individuals actively engaged in health decisions [50]. Person-centered nursing strives to highlight the unique nature of each individual, which lends to their human potential [50]. SD empowers individuals to participate in their own health and educational decisions, ensuring they direct their own physical, psychosocial, and spiritual care [50,51,52]. Nursing’s salutogenic nature supports SD, fostering autonomy, competency, and connectedness by helping patients learn how to manage health challenges and adapt their lifestyle to promote holistic wellbeing within the safety of a trusting nurse–patient relationship [52].

### 3.2. Identification of Attributes, Antecedents, and Consequences

#### 3.2.1. Attributes

According to Rodgers [29,30], the identification of attributes is an important step in clarifying a concept. Attributes associated with the concept of SD are volition, intrinsic motivation, and decision making. Arguably, the most important attribute of SD is intrinsic motivation [23]. Intrinsic motivation occurs when an individual is driven to make a choice for its own sake rather than to gain an external reward [22,23]. Psychological well-being can be enhanced when individuals attain intrinsic goals, whereas succeeding at extrinsic goals provides fewer long-term benefits [51,53,54]. Other attributes of SD are volition and decision making. Volition refers to the power of using one’s own will, also known as “free will” [22,55]. Volition is associated with SD because when individuals practice self-determined behavior, they are acting on their own willpower to do so [21,22]. The last attribute of SD is decision making, which can be defined as the act of making choices. The act of making an independent decision is an essential part of being self-determined [35,39,42]. Individuals demonstrate SD by using their power of free will to make intrinsically motivated decisions. The model case is a real-life example that includes all the identified attributes of the concept [56]. An exemplar is described in Box 1 below.

Box 1Model Case (Attributes are italicized).Joe is a 19-year-old second-year nursing student living on campus at the local University. When the COVID-19 pandemic hit, Joe was forced to live off campus for several months. He is now allowed to live on campus again, but his roommate decided to live off campus, so now Joe lives alone in a private dorm room. He can manage his daily activities, such as laundry, hygiene, picking up his meals from the cafeteria, and schoolwork without assistance. His basic needs are reasonably well met. Although Joe misses his roommate, he has adjusted to living alone. Joe has made friends on campus through his classes, despite being mostly online. When Joe has free time, he enjoys attending socially distanced gatherings on campus and he works as a peer mentor for a first-year nursing student. Joe enjoys the independence of living on campus and he recently *made the decision* to enroll in an elective art class on his own *volition*. Although Joe understands he is not required by the institution to take an art elective, he has always been personally interested in art (*intrinsically motivated*), so he registered himself for the class.

#### 3.2.2. Antecedents

In addition to attribute identification, antecedents and consequences can further clarify a concept, especially one that is context dependent [29]. The general antecedents of SD are free will, having one’s basic needs met, and mental competency. Free will is another antecedent because one cannot act on their own volition without having the unconstrained ability to do so [22,57]. Additionally, one must have their basic needs met (i.e., shelter, water, safety) prior to making self-determined choices [17,42]. For postsecondary students, basic needs during mandatory online learning were internet connection and access to a computer [19]. For example, if a postsecondary student is not able to meet their basic needs, there is little likelihood of them making intrinsically motivated decisions. The last antecedent is mental competency, which is the process of being able to comprehend information to make decisions [58]. If an individual does not possess this antecedent, they are unlikely to effectively take part in the decision-making process [59], thus SD is not plausible.

Within the current context of SD in postsecondary students, additional antecedents should be recognized, such as enrolling in higher education and the COVID-19 pandemic itself. The COVID-19 pandemic is an important antecedent, since research supports that psychological distress in postsecondary students can occur due to the pandemic [20,60]. SD has been shown to decrease psychological distress, which makes the analysis of this concept integral to this population [5,18,19,21]. The act of enrolling in and attending college is also a stipulation. Without these two antecedents, this concept cannot be applied in the context of postsecondary students during the pandemic.

#### 3.2.3. Consequences

Both positive and negative consequences can be outcomes of SD in postsecondary students amid the COVID-19 pandemic. Individuals who practice SD experience a higher level of personal growth, needs satisfaction, and overall well-being than their counterparts [22,23]. According to Ryan and Deci [22,23], when our life activities and choices are congruent with our intrinsic values and motivations, they enable us to reach our full human potential and live authentically. Personal growth through attainment of intrinsic goals is a positive consequence of SD, which is correlated with psychological well-being [23,46,54]. Meeting one’s full and authentic potential can impact an individual’s psychological, physical, and spiritual well-being, supporting the impact of SD on holistic health [61]. As such, postsecondary students who exercise SD may have decreased psychological distress related to the pandemic, as well as increased psychological resilience [62,63]. Thus, postsecondary students who invest in practices that build resilience have positive mental outcomes and decreased COVID-19-related psychological distress [21,64].

Schwinger et al. and Šakan et al. found that adults, in Germany and Serbia respectively, who were unable to practice SD, due to decreased mental functioning and increased stress related to the pandemic, experienced negative consequences [5,65]. This finding may apply to postsecondary students as well [19]. Postsecondary students who do not exercise SD are at risk for dependency on others, such as their parents, teachers, and peers [22,26]. Similarly, postsecondary students may feel disempowered, dissatisfied, and unmotivated due to the COVID-19-related restrictions on daily life [9,18,19,20,66,67]. Many of the daily decisions are being made for postsecondary students by institutional administrators, which may cause students to experience paternalism and lack of autonomy [19,67].

### 3.3. Related Concepts

According to Rodgers [29,30], each concept has related terms that are similar to the concept but are not the same. SD has several related concepts such as independence, control, and autonomy. Independence is the freedom from the control of others [68], whereas control refers to power or influence over events, behaviors, and individuals [69]. Although both terms share characteristics with SD, they cannot be used interchangeably [22,35,42]. Independence differs from SD in that one does not need to possess total independence to be self-determined; for example, an adolescent living at home may depend on their parents for food and shelter or their parents may control their curfew, but they can make self-determined decisions. Control is the possession of power over someone or something outside of oneself [69]. This distinguishes it from SD, which deals primarily with inner self-control [22].

Autonomy has been used as a synonym for SD throughout the literature, but analysis shows that these two terms are related concepts, not surrogates [22,41,42]. Ryan and Deci described autonomy as an “internal perceived locus of causality,” meaning that individuals believe that they control their own decisions [22] (p. 70). Likewise, Ballou stated that autonomy does not originate from an external set of conditions but rather through internal human qualities and beliefs [41]. Albeit similar, SD is less concerned with the individual’s perception of causality, and instead, with acting on one’s right to make intrinsically motivated decisions [22,35,42,54]. In other words, autonomy is the belief that an individual drives their own decision making, whereas SD is the act of making autonomous decisions. These two concepts coexist within SDT but are not surrogates. Therefore, SD in postsecondary students can be defined as the practice or process of making choices without external influence. Figure 1 illustrates a conceptual map of SD from a visual perspective.

## 4. Discussion

This concept analysis has identified the antecedents, attributes, and consequences of self-determination. Table 1 presents the conceptualization of SD through historical evidence as well as the context of culture and population. Although some terms have not been as well documented in the COVID-19-related research, the attributes, antecedents, consequences, and related concepts associated with SD can be noted through different countries and populations. The most common research has been completed in central European countries, such as Germany and Austria, with Germany being the most common (n = 3). Although the foundations of SDT are rooted in the United States, none of the current studies took place there, further supporting the global importance of this topic and possible research gap. In addition, most studies contained a mixed population of postsecondary students, which lends the potential application of SD to a variety of college disciplines and majors. Despite the infancy of this area of research, these data allude to a strong tie between the historical implications of SD and their current application in postsecondary students during the COVID-19 pandemic. Given that the last concept analysis on SD was published in 2014 [42], the need for continued clarification and development through this particular context is warranted.

In the context of the COVID-19 pandemic, SD can be applied to basic needs satisfaction and mental well-being of postsecondary students. In the past, SDT has been applied as a theoretical framework in family psychology and education [22,31]. To justify the selected population for this concept analysis, the application of SD to parenting warrants some discussion. Research shows that when parents become overly involved with their children in a controlling way, this parenting style limits their child’s autonomy and decreases their ability to be self-determined [32,67]. Although these overparenting strategies are meant to promote success and shield children from adversity, they can actually cause young children to experience psychological distress [74]. To meet their basic psychological needs, children must experience a growing degree of autonomy as they age in order to develop their own intrinsic values [26,67]. The notion of limited autonomy is a hallmark feature of over-parenting experienced by individuals considered to be Gen Z [27]. Unlike children, postsecondary students face additional stressors such as living outside their childhood homes independently for the first time. Coming from a highly structured home environment may lead these individuals to experience significant discomfort in unfamiliar situations, such as living on a college campus or in nearby housing [26].

Helicopter and snowplow parenting are characterized by controlling parenting styles that shield children from hardships and adversity [26,67,75,76]. Research has shown that children must face adversity to build resilience [74]. The term helicopter parent describes a parent who “hovers” over a child [77]. This parenting style has gained notoriety in the public media due to the negative effects on postsecondary students [77,78]. Research suggests that helicopter parenting can occur in up to 60% of postsecondary students, regardless of social, economic, and racial and ethnic backgrounds [77,79].

Although helicopter parents are controlling, another parenting style, called snowplow parenting can be equally or more detrimental to postsecondary students [27]. A snowplow parent does not allow their child to fail by “snowplowing” barriers in their lives. These parents can be seen performing tasks like waking up early to ensure their child does not miss their own alarm for school. This child has not been given the opportunity to “fail” (by missing their alarm) because the parent has made sure their child succeeds, thus avoiding any stress or adversity [27]. The more that children and adolescents are shielded by their parents from decision making, the less likely they are to become resilient and self-determined [22,23]. Many postsecondary students are already coming from households where they have an “external perceived locus of causality,” meaning that they are used to their caregiver(s) making decisions for them [22] (p. 70).

Enrolling in higher education should be a time of independence and self-discovery for the postsecondary population. However, the current pandemic-related restrictions imposed by many educational institutions have impacted the ability of these students to exercise autonomy, leading to autonomy frustration [9,20]. When faced with the same autonomy-limited environment at college, this lack of SD can result in psychosocial hardships because students cannot experience their full authentic human potential [61]. Fostering personal growth in this population requires researchers to take a holistic view that includes not only the physical and academic impacts of COVID-19, but also psychological needs [73,80]. Research shows that meeting students’ psychological needs is just as important as maintaining their physical health during periods of uncertainty, such as a pandemic [73,81].

Emerging longitudinal research has shown that COVID-19-related psychological distress is a persistent and lasting problem [82,83]. Hamza et al. [82] found that postsecondary students without preexisting mental illness reported worse mental health over time when compared with students with baseline mental illness; this finding was supported by Lopez Steinmetz and colleagues [83]. These results imply that COVID-19-related psychological distress is a problem in all postsecondary students, regardless of their psychological history. SD has been shown to promote psychological well-being as well as basic needs satisfaction, making it a possible strategy for combating COVID-19-related psychological distress. If SD is encouraged, postsecondary students can potentially experience less psychological distress and a higher quality of life.

The pandemic has caused college administrators to oversee many of the decisions for this population; therefore, it is important that administrators consider the possible influence of SDT on creating and revising campus policies. Teuber et al. [20] found that students who felt engaged and supported by their academic institution during the pandemic were more likely to experience psychological needs satisfaction and less likely to report intent to drop out. Furthermore, Martinek et al. [9] and Eberle et al. [19] discussed the implications of technology accessibility in supporting students’ need for autonomy, relatedness, and competence. For example, students who were unable to access campus resources due to technological limitations experienced psychological needs frustration. Therefore, to support SD, college administrators should ensure that students not only have access to necessary equipment to be involved in the online learning environment but should practice timely communication of policy changes and seek student feedback [9,19,20]. As Sakan and colleagues suggest, administrators and leaders should promote autonomy during the pandemic by involving individuals in the policy-making process [65]. One way to for college administrators to do this is to provide students with opportunities to become more involved, such as by joining a pandemic-related task force or volunteer at a COVID-19 vaccine clinic. SD can also be promoted in other ways, such as encouraging decision making, goal setting, and self-management. Postsecondary students can be empowered by their faculty to take part in decisions about the structure of their online courses, due dates of assignments, and learning activities [15,21]. Research has shown that all three of the basic needs associated with SDT can be promoted successfully in postsecondary students despite distance learning. Although allowing students to regulate their own learning may be tedious or unfamiliar to educators, these strategies can foster intrinsic learning motivation [21]. As postsecondary students transition into the workforce, psychological distress can continue to threaten their ability to be self-determined and live authentically. This makes it even more critical for healthcare workers, educators, and administrators to provide holistic support to promote physical and mental strength in postsecondary students during the pandemic [84].

Previous research supports that resilience is a beneficial trait in promoting the mental health of postsecondary students across the United States, Asia, Australia, and the United Kingdom [85]. One promising area of research is exploring ways to promote psychological resilience in this population during the COVID-19 pandemic [51,75,86,87]. Psychological resilience refers to a person’s ability to cope positively despite adversity, which is crucial to dealing with stress effectively [88]. Postsecondary students, especially Gen Zers, are at risk of decreased resilience due to lack of adversity, which is a key component of building resilience [27,70,76]. Psychological needs satisfaction may help stimulate resilience despite the stress and uncertainty associated with the pandemic [18,21,81], while the inability to meet psychological needs can hinder formation of resilience and lead to heightened stress [70,73]. Stress does not only affect mental health but can also lead to physical illnesses, which are of clinical importance for nurses and other healthcare workers [70,73]. Self-determination and resilience used in tandem have the potential to encourage psychological and physical well-being in postsecondary students. Tools to promote resilience can support self-determination in postsecondary students. Future studies are needed to investigate the relationship between resilience and psychological distress to promote SD in postsecondary students. Ang and colleagues examined how undergraduate students develop resilience during the COVID-19 pandemic [75]. The participants reported that training classes on time management, positivity, and reflexivity enhanced their resilience, especially in the context of experiential learning [75]. Resilience training that incorporates “real-life” case scenarios to promote decision making could improve both SD and resilience in postsecondary students [75].

Although concept analyses focusing on SD have been previously published [35,42], none have explored the concept of SD with special attention to COVID-19 as it applies to post-secondary students. Despite of research on how SDT applies to motivating learning in this population during the COVID-19 pandemic [51,66,71,72], there is limited information applying this concept to pandemic-related psychological distress. This manuscript adds to the current body of knowledge by making connections between several phenomena that are potentially related to SD and impacted by the pandemic, such as psychological distress, autonomy, and resilience. The evolutionary nature of this analysis may serve as a foundation for future research aimed at exploring SD in different settings and populations.

Limitations of this concept analysis are important to note. Given that the COVID-19 pandemic began in March 2020, the literature surrounding this topic is limited. In addition, the use of certain databases and keywords may have reduced search results. There may be additional relevant literature that this author did not find. Another limitation of this concept analysis is that the articles were restricted to the English language, thus reducing the application of SD within the context of different cultural backgrounds.

## 5. Conclusions

The main purpose of this study was to examine the concept of SD as it relates to postsecondary students amid the COVID-19 pandemic. This analysis is significant because it adds to the knowledge base regarding the evolution, significance, and application of the concept of SD in postsecondary students with special attention to this global health context. The analysis also revealed that the concept of SD will require future clarification due to its complexity and applicability to multiple disciplines. SD has evolved from the Puritans seeking religious freedom to the current application to postsecondary students during a pandemic, displaying its wide range of applications and uses. Further development of this concept should focus on strategies to foster SD in this psychologically vulnerable population. It is crucial that educators, counselors, and healthcare workers strive to support postsecondary students during these trying times; however, it is not enough to solely focus on their physical health. Health care should take a holistic approach by caring for individuals physically, emotionally, mentally, and spiritually. Since achieving basic needs satisfaction impacts various components of health, promoting SD is one way to support postsecondary students emotionally while considering current public health issues such as COVID-19. Although the course of the pandemic is still unclear, strategies facilitating SD will not only immediately benefit postsecondary students but will also support their psychological growth, enabling them to reach their full potential.

## Figures and Tables

**Figure 1 ijerph-19-08545-f001:**
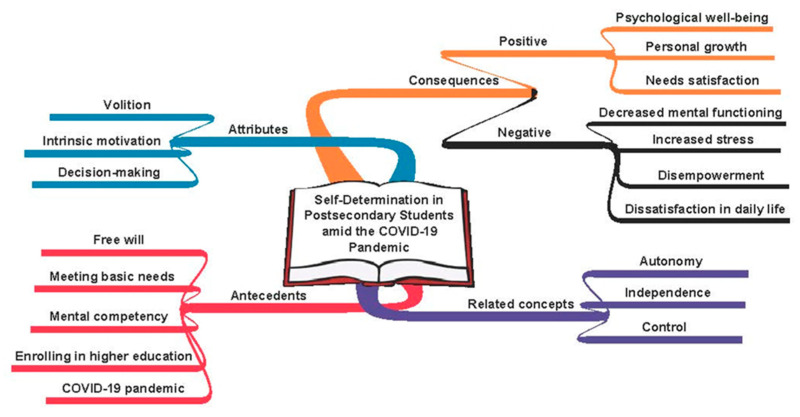
Conceptual Map of Self-Determination.

**Table 1 ijerph-19-08545-t001:** Self-Determination (SD) in postsecondary students.

Conceptualization of SD			
	Historical Evidence of SD (Author, Year)	Application of SD to COVID-19	
		Country	Population
**Attributes**			
Volition Intrinsic Motivation Decision Making	- Deci & Ryan, 1985 [17] - Patient Self-Determination Act, 1990 [38] - Sheldon & Kasser, 1998 [54] - Ryan & Deci, 2000 [22] - Bakitas, 2005 [35] - Weinstein & Ryan, 2011 [70] - Carbonneau et al., 2012 [53] - Ekelund et al., 2014 [42] - American Nurses Association, 2015 [39] - Ryan & Deci, 2020 [23]	- Austria & Finland [21] - Austria & Germany [9] - China [71] - Germany [19,20] - Indonesia [51] - Spain [72] - UK [66]	- Education majors [51] - First-year chemistry students [19] - Medical students [66] - Undergraduate and graduate university students, mixed disciplines [20] - University students, not specified [9,21,71] - Vocational students [72]
**Antecedents**			
Free Will Meeting Basic Needs Mental Competency Enrolling in Higher EducationCOVID-19 Pandemic	- Deci & Ryan, 1985, [17] - Checkland & Silberfield, 1996 [58] - Ryan & Deci, 2000 [22] - Ekelund et al., 2014 [42] - Bakitas, 2005 [35] - Ford, 2010 [59] - Vanskeenkiste et al., 2020 [67] - Vermote et al., 2021 [73]	- Australia [18] - Austria & Finland [21] - Austria & Germany [9] - China [71] - Germany [19,20] - Indonesia [51] - Spain [72] - UK [66]	- Education majors [51] - First-year chemistry students [19] - Medical students [66] - Undergraduate and graduate university students, mixed disciplines [20] - University students, not specified [9,18,21,71] - Vocational students [72]
**Consequences**			
**Positive**			
Psychological Well-being Personal Growth Needs Satisfaction	- Deci & Ryan, 1985 [17] - Sheldon & Kasser, 1998 [54] - Ryan & Deci 2000, 2001, 2020 [22,23,61]	- Australia [18] - Austria & Finland [21] - Austria & Germany [9] - Germany [19,20] - Spain [72]	- First-year chemistry students [19] - Undergraduate and graduate university students, mixed disciplines [20] - University students, not specified [9,18,21] - Vocational students [72]
**Negative**			
Decreased Mental Functioning Increased Stress Disempowerment Dissatisfaction in Daily Life	- Ryan & Deci, 2000 [22] - Weinstein & Ryan, 2011 [70] - Ekelund et al., 2014 [42] - Vanskeenkiste et al., 2020 [67]	- Australia [18] - Austria & Germany [9] - Germany [19,20] - Indonesia [51]	- Education majors [51] - First-year chemistry students [19] - Undergraduate and graduate university students, mixed disciplines [20] - University students, not specified [9,18]
**Related Concepts**			
Autonomy Independence Control	- Deci & Ryan, 1985 [17] - Ballou, 1998 [41] - Sheldon & Kasser, 1998 [54] - Bakitas, 2005 [35] - Ryan & Deci, 2000 [22] - Ekelund et al., 2014 [42] - American Nurses Association, 2015 [39] - Ryan & Deci, 2020 [23]	- Australia [18] - Austria & Finland [21] - China [71] - Germany [19] - UK [66]	- First-year chemistry students [19] - Medical students [66] - University students, not specified [18,21,71]

## Data Availability

Not applicable.

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
