# Peer review of "The Psychological Impact of the COVID-19 Pandemic on Postsecondary Students: An Analysis of Self-Determination"

_ijerph, 2022, doi:10.3390/ijerph19148545_

Round 1

Reviewer 1 Report

I want to thank the authors for their interesting paper.

In general, I do not quite understand how everything fits together. So: “The purpose of the search was to explore how the concept of 88 SD has evolved and to clarify its application to postsecondary students during the 89 COVID-19 pandemic.” Fine, but then the results begin with “SD is a term that appears within several contexts throughout history. Although 136 healthcare is a more recent vehicle for the application of this concept, it can be viewed 137 through an ethical, legal, political, or societal lens. An early example of political SD is the 138 voyage of the Puritans on the Mayflower in 1620 [35,36]. The Puritans sought freedom 139 and ventured to the United States to find solace from religious oppression, finally settling 140 in a colony at Plymouth [35,36]. The exact term SD was first used in a speech by Woodrow 141 Wilson in 1918 when speaking about territorial conquests during World War I [37].”

I think the issue is the goal of the study. How can you do a concept analysis of a concept, if you already ground that concept in a very clear context and population, but then also give a general history?

So, the authors give a general history, beginning at World War 1, but by definition, this is not the concept they are studying, because it is not the concept during COVID-19 and with postsecondary students.

I think it might be safer to say that your concept analysis is either focussed ONLY on the conditions you have stated (COVID-19 and postsecondary students) OR a conceptanalysis of SD, with special attention for COVID-19.

In general, my point is that the aim of the study is not really clear, because do you want to give an insight in the concept, or the concept during COVID-19 among postsecondary students? That divide in your goals is also seen in my opinion in your discussion. You state: “In the context of the COVID-19 pandemic, SD can be applied to basic 283 needs satisfaction and mental well-being of postsecondary students. In the past, SDT has 284 been applied as a theoretical framework in family psychology and education [22,31]. To 285 justify the selected population for this concept analysis, the application of SD to parenting 286 warrants some discussion. Robichaud et al. found that when parents become overly in-287 volved with their children in a controlling way, this parenting style limits their child’s 288 autonomy and decreases their ability to be self-determined [32].”
You begin with “during COVID-19” and then go to “in the past”. Yet, that is not the goal of your study.

I would personally prefer to have a very clear aim in this paper. I imagine there are already hundreds of papers that show that SD originated in the year XXXX during World War 1 et cetera. I think the contribution of your paper should be that you look at purely how it is during COVID-19. Then, in your discussion you can also discuss how it might be different from how it was originally conceptualized.

Young adults between 18 and 24-years-of age, the most common age group 35 for postsecondary students, reported the highest levels of pandemic-related psychological distress when compared to other age groups, with a quarter of them reporting suicidal 37 thoughts [10,11].”The source 11 refers to a university network, no? Because obviously here the largest group will be 18 to 24 year olds, no? I don’t understand what you want to show with reference 11.

“As Florence Nightingale explained, an essential component of the nurse-patient relationship is to assist patients in gaining independence [28].” Strangely, Florence Nightingale is not the author of the source that is referenced. It is odd to explicitly mention her by name. I would remove “As Florence Nightingale explained”, because it seems more like namedropping.

Reviewer 2 Report

The presented manuscript is interesting, but it is necessary to mature more. As the authors and some of their reference indicate, the process is very complex. The studies consulted cover a series of preconceptions and concepts that are very broad, and I suppose that for this reason ranges are opened that are ignored if issues in relation to culture and education have been considered, which are often diverse.

It would be advisable to present one or several tables that allow us to understand how the concepts and their interpretation were crossed. A table with the legal aspects by country, a table of the concepts, to have a map and understand where what is stated in the discussion comes from

Author Response

请参阅附件

Reviewer 3 Report

1.      “Self-determination (SD) is affected by COVID-19-related academic and social restrictions, putting postsecondary students at risk for psychological distress” line 49

The authors have made a strong conviction without any support. This statement should be revised

2.      Rogerian approach line 104, should be Rodgerain?

Round 2

Reviewer 2 Report

I have reviewed the document and observed that the requested adjustments were made. Also, look at the corrections that other reviewers made. I have not further observations.